# Argo salinity: bias and uncertainty evaluation

Annie P. S. Wong[1], John Gilson[2], Cécile Cabanes[3,4]

[1]School of Oceanography, University of Washington, Seattle, WA, United States

[2]Scripps Institution of Oceanography, La Jolla, CA, United States

[3]University of Brest, CNRS, Ifremer, IRD, Laboratoire d'Océanographie Physique et Spatiale (LOPS), IUEM, Brest, France

[4]University of Brest, CNRS, IRD, UAR 3113, IUEM, Brest, France

*Correspondence to*: Annie P. S. Wong (apsw.uw@gmail.com)

**Abstract.** Argo salinity is a key set of in-situ ocean measurements for many scientific applications. However, use of the raw, unadjusted salinity data should be done with caution as they may contain bias from various instrument problems, most significant being from sensor calibration drift in the conductivity cells. For example, inclusion of biased but unadjusted Argo salinity has been shown to lead to spurious results in the global sea level estimates. Argo delayed-mode salinity data are data that have been evaluated and, if needed, adjusted for sensor drift. These delayed-mode data represent an improvement over the raw data because of the reduced bias, the detailed quality control flags, and the provision of uncertainty estimates. Such improvement may help researchers in scientific applications that are sensitive to salinity errors. Both the raw data and the delayed-mode data can be accessed via https://doi.org/10.17882/42182 (Argo, 2022). In this paper, we first describe the Argo delayed-mode process. The bias in the raw salinity data is then analyzed by using the adjustments that have been applied in delayed-mode. There was an increase in salty bias in the raw Argo data beginning around 2015 and peaked in 2017-2018. This salty bias is expected to decrease in the coming years as the underlying manufacturer problem has likely been resolved. The best ways to use Argo data to ensure that the instrument bias is filtered out are then described. Finally, a validation of the Argo delayed-mode salinity dataset is carried out to quantify residual errors and regional variations in uncertainty. These results reinforce the need for continual re-evaluation of this global dataset.

# 1. Introduction

In-situ ocean salinity can be measured accurately by well-calibrated conductivity-temperature-depth (CTD) sensors. By using CTDs mounted on autonomous floats, the global Argo Program has collected over two million vertical profiles of temperature-salinity (*T/S*) versus pressure (*P*) in the past 20 years. Many of these floats receive pre-deployment CTD accuracy checks to ensure that the sensor calibrations are within the manufacturer's specifications. However, over time these sensors can become affected by contamination, or undergo physical changes that alter their accuracy. Recalibration of these CTDs involves retrieval of the floats, which can occur when opportunities arise. However, such retrieval occasions are infrequent and not extensive. To determine if post-deployment adjustment of its data is necessary, Argo uses a set of delayed-mode procedures that makes use of reference data. These Argo delayed-mode salinity data are typically available about 12 to 18 months after the vertical profiles are collected.

Argo data are used in many oceanographic applications, forecasting services, climate research, ocean modeling, and data products. However, using the data without post-deployment adjustment can lead to spurious scientific results. This effect has been shown to be especially impactful when using Argo salinity data collected after 2015, when a higher-than-average number of CTDs on Argo floats developed sensor drift towards higher salinity values (Wong et al., 2020). Ponte et al. (2021) compared estimates of in-situ global mean salinity $\bar{S}$ from 5 different data products that included Argo data. They found a spurious increase in $\bar{S}$ after 2015 in all the products, except the Roemmich and Gilson (2009) climatology. The spurious increase in $\bar{S}$ after 2015 was postulated to be the result of inclusion of biased Argo salinity data that have not been adjusted in delayed-mode, while the absence of this artificial increase in $\bar{S}$ in Roemmich and Gilson (2009) was attributed to stricter quality control of the affected data. Similar discrepancies were seen in comparisons between global ocean mass change (Chen et al., 2020) and global mean sea level budget (Barnoud et al., 2021) derived from GRACE/GRACE-FO and Altimeter-Argo. In both studies, the discrepancies become substantially larger after 2015 and are likely related to using biased but unadjusted Argo salinity.

The Joint Committee for Guides in Metrology (2008) defines *measurement error* as the difference between the measured and the true value of a variable. It has two components: a random component and a systematic component. The random component is influenced by unpredictable effects and cannot be corrected. The systematic component, or bias, arises from recognized effects

and thus can be corrected. When all the components of error have been evaluated and corrected,
*uncertainty* refers to the doubt about the validity of the evaluation and the correction. Quantifying
the uncertainties of an ocean dataset increases its usefulness to scientists and other stakeholders
(Elipot et al., 2022).

The instruments used in Argo and the impacts that their respective technical limitations

have on the data have been described in Wong et al. (2020). The uncertainties of Argo data have
been assessed by comparison with high-quality shipboard measurements, and are concluded to be
near the manufacturer instrument accuracy specifications of 0.002°C for temperature and 2.4 dbar
for pressure. For salinity, even though the manufacturer specified initial instrument accuracy is
0.0035 psu (0.0003 Siemens per meter at 2°C and 2000 dbar), the uncertainties of Argo salinity
have been assessed to be around 0.01 psu (Riser et al., 2008; Wong et al., 2020).

This paper aims to improve understanding of the treatment and uncertainty of Argo salinity

data. Section 2 describes the evolution of Argo's salinity adjustment method and its
implementation. Section 3 describes the temporal and spatial distribution of bias in the raw Argo
salinity. The best ways to use Argo data are described in Sect. 4. Lastly, an evaluation of the
uncertainty in Argo's delayed-mode salinity data against a shipboard CTD reference database is
discussed in Sect. 5.

## 2. Argo salinity adjustment method and implementation


### 2.1. Argo's salinity adjustment method

Measurement stability refers to an instrument's ability to repeat the same measurement over time.
The change in the instrument's bias over time is referred to as sensor drift. A system for adjusting
sensor drift in Argo salinity data was originally developed by Wong et al. (2003). The system uses
an objective mapping technique to estimate the background salinity field along the trajectory of
each float. Mapping is done on a set of fixed θ surfaces and relies on nearby reference data. Salinity
data from each float are fitted to the objectively mapped field in potential conductivity space by
weighted least squares. The time-varying component is smoothed out by another least squares fit
over multiple profiles to filter out the transient oceanic noise in the float data and the reference
data. The result is a multiplicative correction in conductivity, or an additive correction in salinity,
for each vertical profile. Böhme and Send (2005) improved on the original method by using float-
observed θ surfaces and introduced potential vorticity as a factor for selecting reference data in
areas affected by topographic constraints. Owens and Wong (2009) combined the original method
with the improvements of Böhme and Send (2005) and introduced a piecewise linear fit with the
Akaike Information Criteria in the treatment of the time series. Moreover, the analysis was done
on 10 best float-observed θ surfaces that had minimum salinity variance. More recently, Cabanes
et al. (2016) suggested modifications to better account for interannual variability and provide more
realistic error estimates.

As these methods evolve, their authors have maintained a set of computational code that
can be used by all Argo float providers. Transparency and reproducibility of the salinity
adjustments are achieved via this provision of code that operates on the raw measurement inputs
to produce the delayed-mode adjusted data. Currently, the code used for salinity adjustment in
Argo is a combined set from Owens and Wong (2009) and Cabanes et al. (2016). See
github.com/ArgoDMQC/matlab_owc.

These salinity adjustment methods rely on accurate reference data. To that end, two
reference databases are provided internally in Argo for salinity adjustment: 1. a reference database
which consists of shipboard CTD data (internally named CTD_for_DMQC, maintained by
Coriolis Data Center), and 2. a reference database which consists of Argo data that have been
verified as having good quality without needing adjustments (internally named Argo_for_DMQC,
maintained by Scripps Institution of Oceanography). These two reference databases are updated
approximately once a year to account for the constantly changing oceans.

## 2.2. How is salinity adjustment implemented in Argo?


Delayed-mode salinity evaluation in Argo is carried out by each data-providing group, and not by
a central institution. Each data-providing group in Argo has a team of delayed-mode operators who
manually inspect the data. As both pressure and temperature are required to measure salinity, all 3
parameters ($P, T, S$) are evaluated together in delayed-mode. Random point-wise errors, such as
spikes, are flagged as bad data. Sensor drifts are identified and either adjusted or flagged as
unadjustable data. Evaluation of sensor drifts, not to be confused with real ocean signals, requires
significant oceanographic knowledge, scientific judgment, and insights based on experience. To
ensure all data-providing groups are consistent in following best practices, two technical
documents are maintained internally in Argo to describe the data processing procedures and to
provide examples. These are: 1. Argo Quality Control Manual for CTD and Trajectory data (Wong
et al., 2022), and 2. DMQC Cookbook for core Argo parameters (Cabanes et al., 2021). These are
living documents, modified and updated as the data processing procedures develop and evolve.

Due to the need to accumulate a time series for reliable evaluation of sensor drifts, delayed-

mode data for a float may not be available until a sufficiently long time series from that float has
been accumulated. The timeframe for availability of delayed-mode data is therefore dependent on
the nature of the sensor drift, as well as the availability of the delayed-mode operators. In general,
most Argo delayed-mode salinity data are available about 12–18 months after the raw
measurements are collected. These data are re-evaluated periodically to reduce inconsistencies
between the various data-providing groups. Therefore, Argo delayed-mode data are "dynamic"
data that continually change and improve over time.

## 3. Bias in Argo raw salinity data
Bias in raw Argo salinity can contain effects from three different sources:
1. error from the pressure measurements (Barker et al., 2011);
2. error from conductivity cell thermal inertia, due to the lag between the temperature and
conductivity measurements (Johnson et al., 2007; Martini et al., 2019; Dever et al., 2022);
3. error from conductivity cell sensor drift (Wong et al., 2020).

The effect of pressure error on salinity is not negligible. For example, assuming standard

seawater properties of $S = 35$ and $T = 15°C$, a pressure error of 10 dbar will result in a salinity error
of about 0.004 psu. However, less than 1% of Argo vertical profiles have identifiable pressure
error of greater than 10 dbar. The effect of the conductivity cell thermal inertia error on salinity
can exceed 0.01 psu in regions of strong temperature gradients, such as the base of the mixed layer,
but is negligible (<0.002 psu) elsewhere.

The bias caused by conductivity cell sensor drift is the most significant error in Argo

salinity. Some of this bias cannot be corrected, as severe sensor drift (and other CTD malfunctions)
can cause data corruption that is beyond salvage. The remaining adjustable bias, $\partial S$, can be
estimated by using the salinity adjustments that have been applied in delayed-mode:

$$\partial S = \overline{Sraw - Sadjusted}$$

where *Sraw* are the raw Argo measurements and *Sadjusted* are the corresponding delayed-mode adjusted values. Here, we compute $\partial S$ for each Argo vertical profile that has delayed-mode adjusted data, but only use measurements deeper than 600 dbar to exclude the effects of the cell thermal inertia error. Profiles with identifiable pressure error greater than 10 dbar ($|\overline{Praw - Padjusted}| > 10\ dbar$) are excluded to factor out the effects of pressure error on salinity. We consider the profiles with $|\partial S| < 0.002$ as good data that have not been affected significantly by sensor drift. Thus, the remaining $\partial S$ represents the typical bias magnitude identified mostly from conductivity cell sensor drift. Here, a positive $\partial S$ means the raw values are higher than true, or drifted towards saltier values (salty drift). Similarly, a negative $\partial S$ means the raw values are lower than true, or drifted towards fresher values (fresh drift).

Salty drift is the dominant mode of sensor drift in Argo salinity, with about 10% of all Argo profiles having a positive adjustable bias (Fig. 1a, blue bars). Most of the physical causes of salty drift are unknown. One known cause was determined to be due to the early deterioration of the encapsulant material in CTDs manufactured by Sea-Bird Scientific starting in 2015. Changes at the manufacturing level were introduced in 2018 to reduce such occurrences. The number of Argo profiles with adjustable salty drift increased steadily from 2000 and peaked in 2017-2018 at about 17% of the annual profiles count. This 2017-2018 peak (Fig. 1a), as well as the annual average of adjustable bias (Fig. 1b), may shift slightly as more delayed-mode evaluated profiles become available in the future, but the present result is consistent with the timeline of the CTD encapsulant issue.

On the other hand, fresh drift occurred more frequently in the early years of Argo (Fig. 1a, red bars), reaching a peak of about 28% of annual profile count in 2001-2002. The subsequent decline is broadly coincident with the introduction of Iridium in 2005 for data communication. Fresh drifts are mostly caused by contamination of the CTD while the floats remain at the sea surface for communication with satellites. Earlier floats that used the ARGOS System, which was the predominant telecommunication system before Iridium, typically spent between 6 to 18 hours at the sea surface for data telemetry. With Iridium, the time spent at the sea surface is reduced to about 30 minutes, thus reducing the risk of CTD contamination. The number of Argo profiles with adjustable fresh drift accounts for about 4% of all Argo profiles.

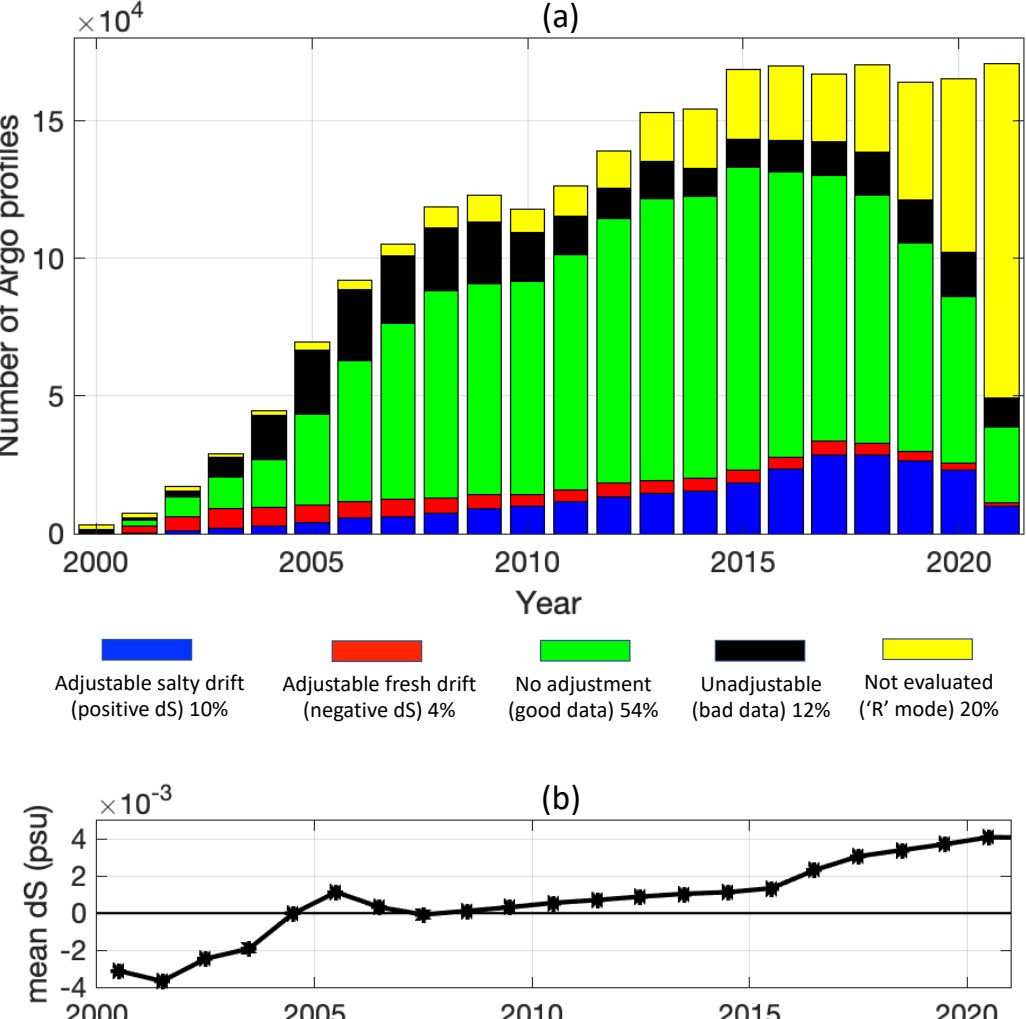

Figure 1: (a) Temporal distribution of Argo salinity delayed-mode evaluation. Values are from April 2022. (b) Annual average of all delayed-mode salinity adjustments, which is an estimate of the adjustable bias in the raw Argo salinity data.

The magnitude of adjustable bias can be an indicator of sensor limitation. Amongst all the salinity profiles with adjustable sensor drift, salty or fresh, about 90% have magnitude < 0.03 (Fig. 2). Only 2-3% of adjustable sensor drift have magnitude > 0.05. Some of the larger-magnitude adjustments were concentrated in the Atlantic and the North Pacific in the early years of Argo before 2010 (Fig. 3), when delayed-mode efforts were focused in those areas that had more reference data, and when delayed-mode operators had less experience evaluating larger-magnitude

adjustments. Indeed, beyond the 0.05 limit, salinity data with sensor drift usually show signs of
unrecoverable damage, and applying such large adjustments to the exceptional cases should only
be done with sound judgement. For the unrecoverable profiles, no adjustment is applied, and the
data are flagged as bad in the Argo data files (Wong et al., 2022). These unadjustable salinity data,
plus those corrupted by other CTD or float malfunctions, account for about 12% of all Argo
profiles. As of time of analysis, about 54% of Argo profiles were considered to be of good quality
and with no identifiable bias, and about 20% of Argo profiles remained in waiting for delayed-
mode evaluation.

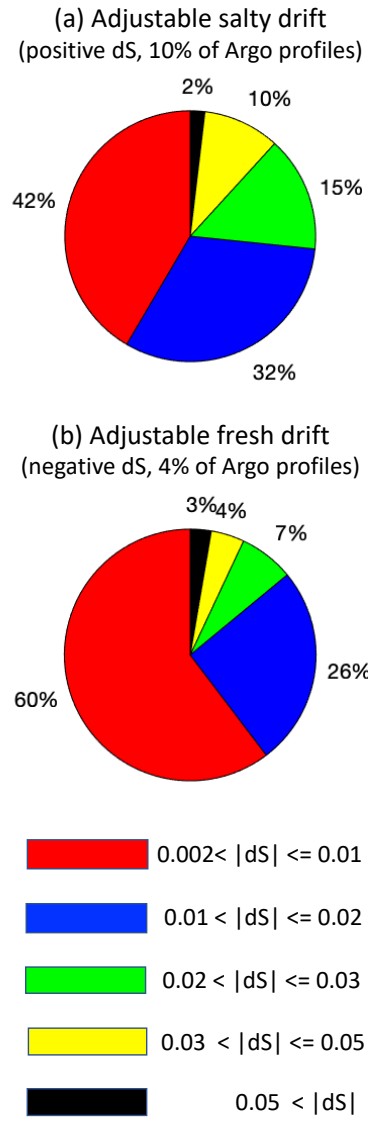


Figure 2: Magnitude of Argo delayed-mode salinity adjustments, as of April 2022. (a) Adjustable salty drift. (b) Adjustable fresh drift.

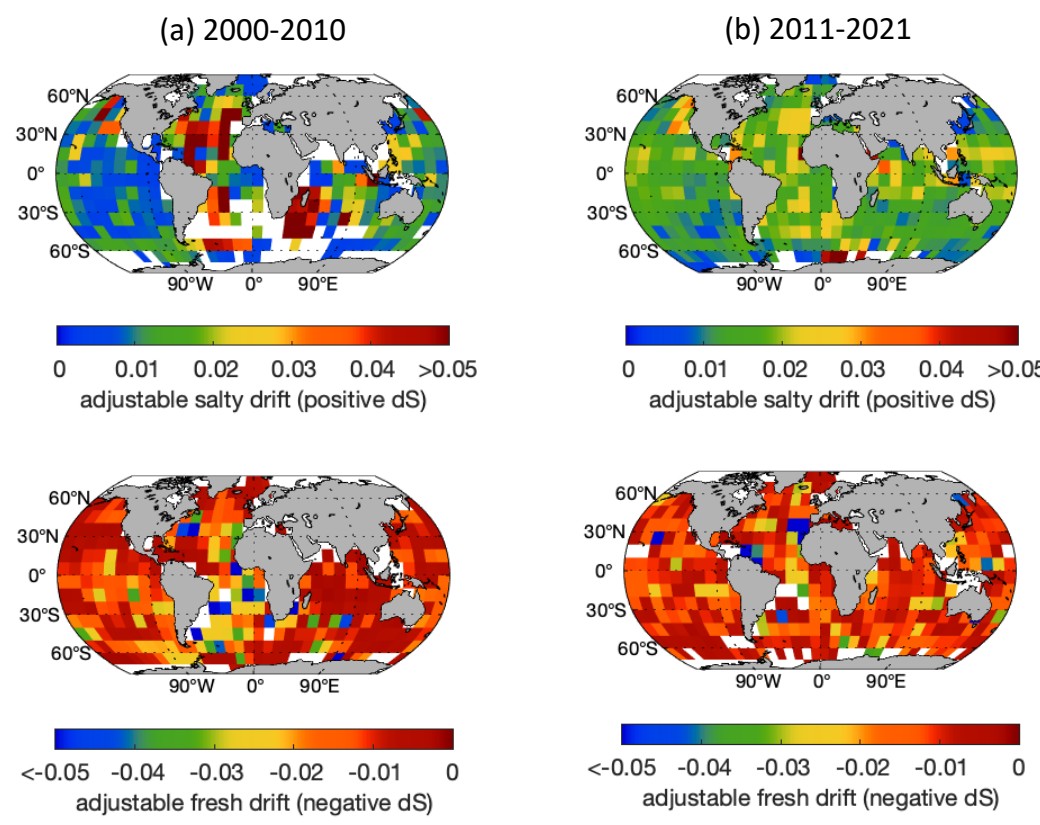

Figure 3: Spatial distribution of Argo delayed-mode salinity adjustments, as of April 2022. (a) 2000-2010. (b) 2011-2021. Top panels show adjustable salty drift (positive dS). Bottom panels show adjustable fresh drift (negative dS). Colors indicate the mean of dS in each 10°×10° grid square. White color denotes areas with no Argo data or no appropriate dS at the time of this analysis.

## 4. How to use Argo data: raw data, adjusted data, data products

In all the Argo data files, parameter values are stored in two variables: PARAM and PARAM_ADJUSTED. Data from the CTDs are stored in PARAM = PRES, TEMP, PSAL. For

biogeochemical data, please refer to Bittig et al. (2019). The PARAM variables store the original
raw measurements, while the PARAM_ADJUSTED variables store the corresponding
evaluated/adjusted values. Both the raw data and the corresponding evaluated/adjusted data are
available in the same Argo data files as a practice of good data stewardship. Since the
evaluated/adjusted data are based on the original raw measurements, archival of the original raw
measurements are important to allow checking of the data processing procedures. Therefore, the
raw data are preserved as originally received, to serve as a record if questions arise later.

Argo data files that contain data evaluated/adjusted in delayed-mode are denoted by
DATA_MODE = 'D'. Some Argo data centers can extract the most recent delayed-mode salinity
adjustment and apply it to later, newly collected profiles in real-time. This procedure can provide
intermediate-quality salinity data to users in real-time, and the data files are denoted by
DATA_MODE = 'A'. When neither delayed-mode nor real-time adjustment is available, only the
raw data are available, and the data files are denoted by DATA_MODE = 'R'. Figure 4 illustrates
the general meaning of these variables. Each data point, raw and evaluated/adjusted, has an
associated quality control flag (PARAM_QC and PARAM_ADJUSTED_QC) that provides
qualitative assessment of the value (Table 1). In addition, each delayed-mode evaluated/adjusted
data point has an associated variable, PARAM_ADJUSTED_ERROR, that records the
quantitative uncertainty of the evaluated/adjusted value. Scientific users should use the
evaluated/adjusted values in PARAM_ADJUSTED, together with their QC flags in
PARAM_ADJUSTED_QC and uncertainty values in PARAM_ADJUSTED_ERROR, whenever
possible. The highest quality data are obtained by selecting PARAM_ADJUSTED with
PARAM_ADJUSTED_QC = '1' and DATA_MODE = 'D'.

Increasing time needed for data processing

**Real-time data files: available within 12-24 hrs**
**Filename convention: Rwmoid_cyclenumber**

DATA_MODE = 'R' (real-time processing)

PARAM = raw measurement
PARAM_QC = qc flag of raw measurement
PARAM_ADJUSTED = not available
PARAM_ADJUSTED_QC = not available
PARAM_ADJUSTED_ERROR = not available

DATA_MODE = 'A' (adjusted in real-time)

PARAM = raw measurement
PARAM_QC = qc flag of raw measurement
PARAM_ADJUSTED = real-time adjusted value
PARAM_ADJUSTED_QC = qc flag of real-time adjusted value
PARAM_ADJUSTED_ERROR = not available

**Delayed-mode data files: usually available after 12 months**
**Filename convention: Dwmoid_cyclenumber**

DATA_MODE = 'D' (delayed-mode processing)

PARAM = raw measurement
PARAM_QC = qc flag of raw measurement
PARAM_ADJUSTED = delayed-mode adjusted value
PARAM_ADJUSTED_QC = qc flag of delayed-mode adjusted value
PARAM_ADJUSTED_ERROR = uncertainty of delayed-mode adjusted value

Increasing quality of the evaluated/adjusted data

Figure 4: The variables in an Argo data file and their different timeframe of availability. Data from CTDs are stored with PARAM = PRES, TEMP, PSAL. For biogeochemical data, please refer to Bittig et al. (2019). The highest quality Argo data are those stored in PARAM_ADJUSTED, with PARAM_ADJUSTED_QC = '1' and DATA_MODE = 'D' (delayed-mode).

| QC Flag | Meaning | Real-time comment (applicable to <PARAM>_QC in 'R' mode and <PARAM>_ADJUSTED_QC in 'A' mode) | Delayed-mode comment (applicable to <PARAM>_ADJUSTED_QC in 'D' mode) |
|---|---|---|---|
| '0' | No QC is performed | No QC is performed. | No QC is performed. |
| '1' | Good data | Good data. All Argo real-time QC tests passed. These measurements are good within the limits of the Argo real-time QC tests. | Good data. No adjustment is needed, or the adjusted value is statistically consistent with good quality reference data. An error estimate is supplied. |
| '2' | Probably good data | Probably good data. These measurements are to be used with caution. | Probably good data. Delayed-mode evaluation is based on insufficient information. An error estimate is supplied. |

| | | | |
|---|---|---|---|
| '3' | Probably bad data that are potentially adjustable | Probably bad data. These measurements are not to be used without scientific adjustment, e.g. data affected by sensor drift but may be adjusted in delayed-mode. | Probably bad data. An adjustment may (or may not) have been applied, but the value may still be bad. An error estimate is supplied. |
| '4' | Bad data | Bad data. These measurements are not to be used. A flag '4' indicates that a relevant real-time qc test has failed. A flag '4' may also be assigned for bad measurements that are known to be not adjustable, e.g. due to sensor failure. | Bad data. Not adjustable. Adjusted data are replaced by FillValue. |
| '5' | Value changed | Value changed | Value changed |
| '6' | Not used | Not used | Not used |
| '7' | Not used | Not used | Not used |
| '8' | Estimated value | Estimated value (interpolated, extrapolated, or other estimation) | Estimated value (interpolated, extrapolated, or other estimation) |
| '9' | Missing value | Missing value. Data parameter will record FillValue. | Missing value. Data parameter will record FillValue. |
| ' ' | FillValue | Empty space in netcdf file. | Empty space in netcdf file. |


Table 1. Argo quality control (QC) flags. Additional information on these QC flags can be found
in "Notes on the Argo QC flags" in Argo Quality Control Manual for CTD and Trajectory data
(Wong et al., 2022, Section 6.1).

The two Argo Global Data Assembly Centers (Argo GDACs, at Coriolis France and at
FNMOC USA) hold a "grey list", which contains a list of active Argo floats that are suspected of
malfunctioning. This grey list is a means for the Argo real-time data centers to automatically flag
incoming data from suspicious floats with lower-quality QC flags. However, the grey list is not a
comprehensive list of problematic floats, as some malfunctioning floats may not be detected early
enough to be grey-listed, and those that are grey-listed are removed from the list when they become
inactive. Therefore, users should not rely on the Argo grey list alone to filter out bad data, but
should use the QC flags. The most complete information regarding the quality of Argo data is
contained in the Argo QC flags.
Since Argo delayed-mode data can become available at different times and are subject to
revisions, users should refresh their data holdings periodically from the Argo GDACs to obtain
the most recent evaluation and adjustments. There are currently many scientific data products that
include Argo data. However, these data products are not part of the Argo data system and are not
held accountable by Argo. When using scientific data products derived from Argo data, users are
urged to check to what extent raw data are used, what data quality control is done beyond those
provided by Argo, and how often reanalysis is done that includes the most recent Argo delayed-
mode data.

## 5. Uncertainty in Argo delayed-mode salinity data

As described in Sect. 3, Argo delayed-mode salinity data consist of three different evaluation
outcomes:
1. data are considered to be of good quality and contain no identifiable bias, hence no adjustment
is applied;
2. data are considered to be affected by sensor drift that are adjustable, hence adjustments are
applied;
3. data are considered to be bad and unadjustable.
The uncertainty in Argo delayed-mode salinity data is therefore a combination of uncertainties in
the evaluation and in the applied adjustments, both of which are due to incomplete knowledge of
the true value of the measurements. Such is the nature of oceanographic data collected by
autonomous instruments operating without contemporaneous and co-located reference data.
As described in Sect. 4, the highest quality Argo salinity data are those stored in the
variables PSAL_ADJUSTED, with PSAL_ADJUSTED_QC = '1' and DATA_MODE = 'D'
(delayed-mode). Here, we evaluate the uncertainty in these highest quality Argo delayed-mode
salinity data from 2000 to 2021 by comparing them to the shipboard CTD reference database,
CTD_for_DMQC. The CTD_for_DMQC reference database contains data from the World Ocean
Database and GO-SHIP, which are considered the best estimates of the true ocean salinity field.
This same database is also used as part of the Argo delayed-mode salinity evaluation and
adjustments (with some evaluation aided by a second reference database, Argo_for_DMQC).
However, while the Argo delayed-mode process considers data from each float separately, this
analysis considers data from all floats collectively. Moreover, the CTD_for_DMQC reference
database is enriched over time, and may contain more data today than when the delayed-mode
evaluation was done. We do note that this analysis may not satisfy the standard of a rigorous
regression validation, where a completely independent dataset is needed. Nonetheless it provides
a means to examine the uncertainties in the global Argo salinity dataset.

This analysis was focused on Argo profiles that extended to 2000 dbar. Additional visual

inspection was done on the delayed-mode salinity profiles to remove gross outliers that remained.
These were generally contaminated profiles that had not been adjusted or flagged properly, and
amounted to <1% of the delayed-mode dataset as of the time of this analysis. The remaining Argo
delayed-mode profiles and reference CTD profiles were grouped into grid squares of 10° latitude
by 10° longitude. In each square, an isotherm with relatively uniform salinity (small salinity
variance) was selected. In the upper 2000 dbar of the world's oceans, this isotherm is usually at
>1000 dbar. But in regions where there is a confluence of multiple water masses at >1000 dbar,
this isotherm can be from shallower pressures (Owens and Wong, 2009). For example, in the
subtropical South Atlantic, Upper Circumpolar Water overrides the warmer but saltier Upper
North Atlantic Deep Water, thus creating a slight temperature inversion at around 1600 dbar
(Mémery et al. 2000). Hence the isotherm with lesser salinity variance in the subtropical South
Atlantic is in the mode water or central water pressure range of 400-1000 dbar. Comparison of
salinity is better done on isotherms than on isobars, because differences on isobars can contain
effects of the vertical movement of isotherms over time.

In each square, each Argo delayed-mode profile was compared against the nearest

reference CTD profile within a 3° radius circle and 15 years of age. Argo/refCTD salinity
difference, $\Delta S_{\text{Argo-refCTD}}$, was then computed for each Argo/refCTD pair on the selected isotherm
in that square. This comparison method is limited by the spatial and temporal availability of the
reference CTD data. For example, with the search criteria of 3° radius circle and 15 years of age,
only about 20% of Argo delayed-mode profiles had nearby reference CTD profiles with which to
compare at the time of this analysis. The comparison results will contain effects of spatial and
temporal variabilities of the water masses, but these are minimized by using isotherms with
relatively uniform salinity.

The statistical distribution of $\Delta S_{\text{Argo-refCTD}}$ provides a measure of the overall uncertainty

(Fig. 5). The mean and the median of the distribution of $\Delta S_{\text{Argo-refCTD}}$ are at approximately 0 (mean
= −0.0003, median = −0.0007), with the standard deviation σ = 0.017. This means the Argo
delayed-mode salinity data selected in this comparison agree with nearby reference CTD data on
average. About 64% of $\Delta S_{\text{Argo-refCTD}}$ are within ±0.01.

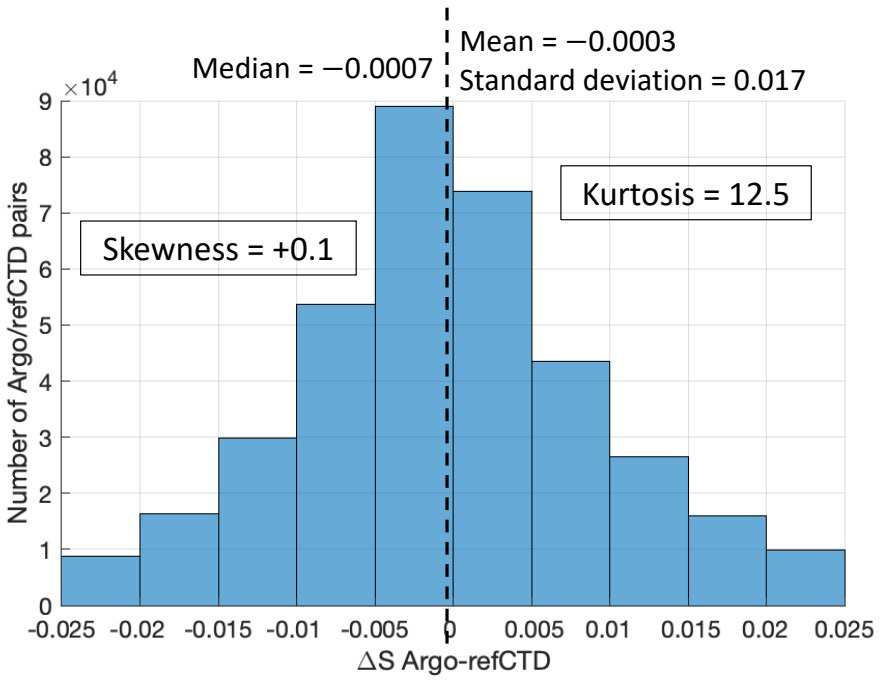


Figure 5: Statistical distribution of $\Delta S_{\text{Argo-refCTD}}$, as of April 2022. The Argo data used in this
analysis are delayed-mode salinity data from PSAL_ADJUSTED, with PSAL_ADJUSTED_QC
= '1' and DATA_MODE = 'D'. Note that this analysis only accounts for about 20% of the Argo
delayed-mode salinity data. For comparison, a normal distribution has skewness = 0 and kurtosis
= 3.

The kurtosis of the statistical distribution of $\Delta S_{\text{Argo-refCTD}}$ is 12.5. Kurtosis is a measure of
the heaviness of the tails of a distribution, or, how large the outliers are. (For comparison, a normal
distribution has a kurtosis of 3). About 18% of $\Delta S_{\text{Argo-refCTD}}$ are outside the range of ±0.017 (±1σ).
These are regions with higher uncertainties in delayed-mode evaluation (Fig. 6), due to either
inadequate reference CTD data, or higher regional salinity variability, or both. The main high-
uncertainty regions are the western Indian Ocean, the subtropical North and South Atlantic Ocean,
and other near-coast areas that are influenced by coastal processes. The Southern Ocean does not
show up as a high uncertainty region in this analysis because Circumpolar Deep Water, which is a
water mass in the Southern Ocean with relatively uniform salinity, usually provides robust results
in delayed-mode analysis. Overall, these uncertainties can be reduced if more contemporaneous
and co-located reference CTD data are available for delayed-mode analysis. These can be bottle-
calibrated CTD casts from deployment, or from research cruises that sample regions not covered
by GO-SHIP.
The statistical distribution of $\Delta S_{\text{Argo-refCTD}}$ is slightly skewed to the fresh side (skewness =
+0.1). Skewness is a measure of the asymmetry of the distribution, with positive skewness meaning
a longer tail on the positive side, or, that the distribution leans more to the negative (fresh) side.
Figure 6 shows that the Argo delayed-mode profiles that are slightly fresher than reference CTD
data are mostly located in the equatorial band 10°S to 10°N in the Pacific and Atlantic oceans, and
in the circumpolar Southern Ocean south of 60°S. The selected isotherms for estimating $\Delta S_{\text{Argo-}}$
$_{\text{refCTD}}$ typically have potential density anomalies $\sigma_0 > 27.6$ kg m$^{-3}$ in the equatorial Pacific, $> 27.7$
in the equatorial Atlantic, and $> 27.8$ south of 60°S. Hence these are deep water masses that do not
show much decadal change. We speculate that this minor fresh skewness is instrument noise that
has remained in the Argo delayed-mode dataset. During delayed-mode evaluation, it is often easier
to identify strong sensor drifts than mild instrument calibration offsets, as the latter requires
verification from contemporaneous, co-located reference data, which are often lacking. It is
therefore possible that many mild instrument offsets, fresh or salty, have not been adjusted. The
residual fresh bias is more apparent in regions such as the equatorial Pacific and Atlantic, where
the deep T/S relations allow for easier delayed-mode adjustment of sensor drifts, and which then
emphasize the unadjusted fresh offsets. In other regions where delayed-mode evaluation is more
difficult, this residual fresh bias could be masked by the surrounding variability, and so is not as
apparent.

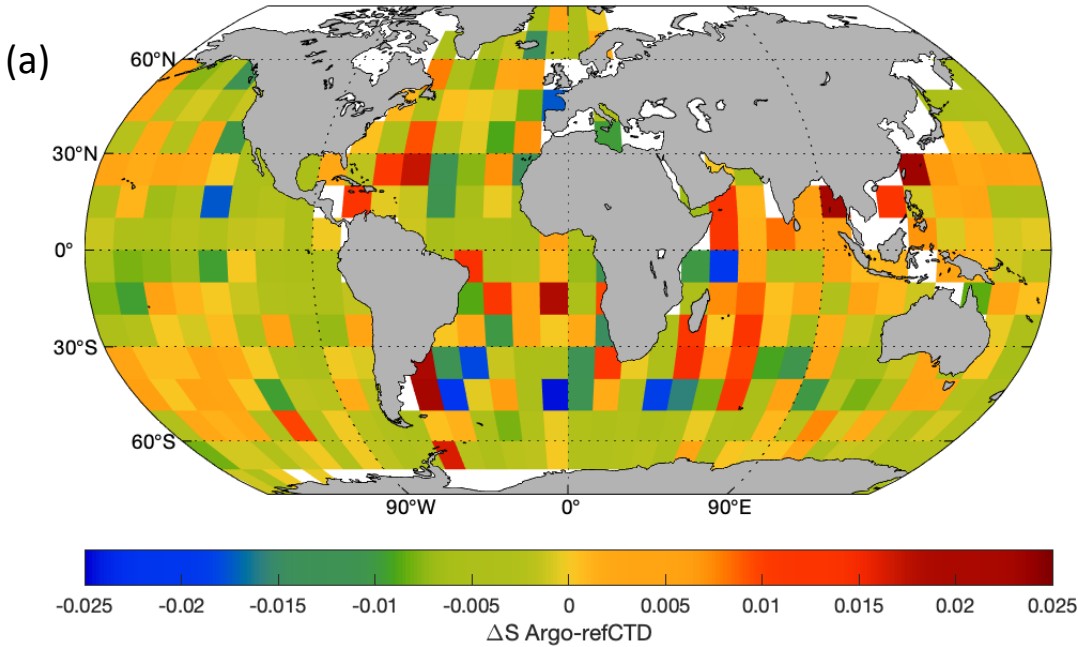

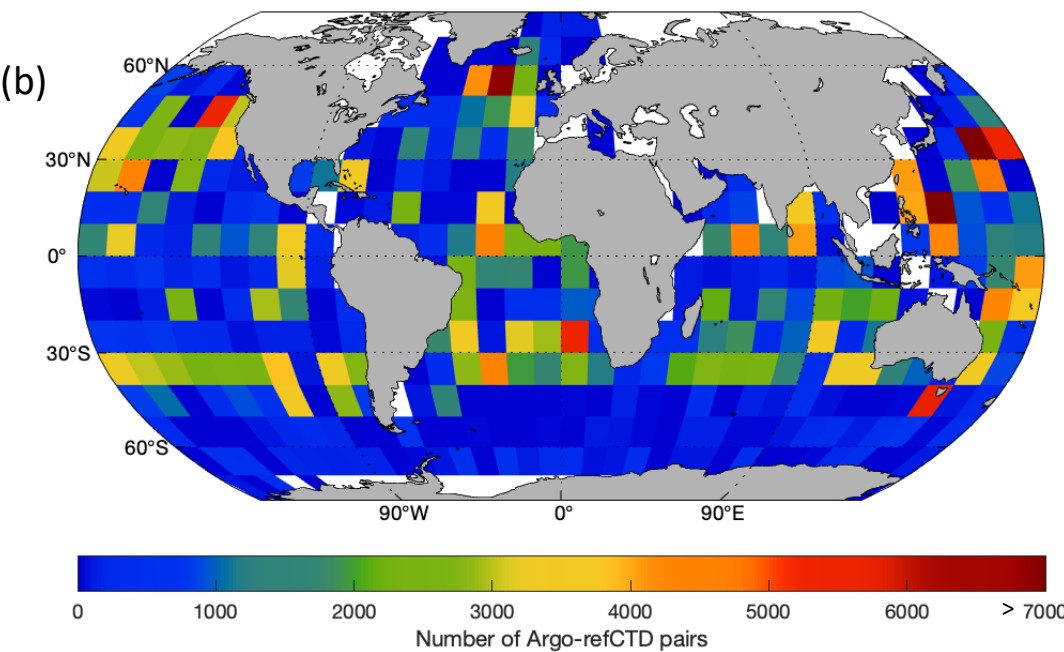



Figure 6: (a) Spatial distribution of $\Delta S_{\text{Argo-refCTD}}$, averaged in 10°x10° grid squares, and (b) number
of Argo-refCTD pairs in each 10°x10° grid square. The Argo data used in this analysis are delayed-
mode salinity data from PSAL_ADJUSTED, with PSAL_ADJUSTED_QC = '1' and
DATA_MODE = 'D', as of April 2022. Note that this analysis only accounts for about 20% of the
Argo delayed-mode salinity data. White color denotes areas with no Argo data or no Argo-refCTD
match at the time of this analysis.


## 6. Discussions and Summary

This paper uses the salinity adjustments that have been applied in delayed-mode to estimate the
bias in the raw, unadjusted Argo salinity data from 2000 to 2021. There is an increase in the annual
average of adjustable bias since 2015, due to the disproportionately high number of salty-drifting
CTDs since 2015. The amount of salinity data that have been declared as bad and unadjustable has
also increased during that period. While Argo salinity data that are adjustable typically have bias
of magnitude < 0.05, those that are unadjustable can have bias with magnitude > 0.05. Inclusion
of these raw biased data in scientific applications, such as gridded ocean salinity products, has
been demonstrated to create spurious results (e.g. Liu at al., 2022).
This salty bias in the raw Argo salinity data is expected to decrease in the coming years as
the underlying manufacturer problem has likely been resolved. We note that even though the
period 2015–2020 saw a large percentage of data loss due to the CTD problem that caused the
increased salty drifts, historically there was a larger percentage of data loss from the period 2004–
2011 (Fig. 1a, black bars). Those earlier CTD failures were partly the results of the Druck
"snowflakes" and the Druck "oil microleak" problems (Wong et al, 2020). These instrument issues
emphasize the importance of improving sensor stability, especially in light of the increase in float
lifetime. As the average lifetime of an Argo float increases, the sensors will be required to spend
more time in the ocean, which will increase the likelihood of sensor drift or malfunction. Hence
sensor reliability needs to be improved to ensure a healthy return of good quality data.
In all Argo data files, both the raw data and the delayed-mode data are available as a
practice of good data stewardship. The delayed-mode data represent an improvement over the raw
data because of the reduced bias, the detailed quality control flags, and the provision of uncertainty
estimates. Scientific applications that are sensitive to salinity errors should therefore use the
delayed-mode data provided by Argo. When accessing data from Argo data files, the highest
quality Argo delayed-mode salinity data are obtained by selecting values in PSAL_ADJUSTED,
with PSAL_ADJUSTED_QC = '1' and DATA_MODE = 'D' (delayed-mode). We analyzed these

highest quality Argo salinity data (as of April 2022) to 2000 dbar against a shipboard CTD reference database to assess their uncertainty. The statistical distribution of $\Delta S_{\text{Argo-refCTD}}$, computed on isotherms with small salinity variance, showed mean and median values close to zero, suggesting good agreement on average between the selected Argo delayed-mode data and nearby reference CTD data. The distribution had a kurtosis of 12.5 and a skewness of +0.1. Hence it is not exactly a normal distribution, which has a kurtosis of 3 and a skewness of 0. We note that such statistics are dependent on sample sizes, and this analysis only accounts for about 20% of all Argo delayed-mode salinity data as of April 2022, being limited by the availability of nearby reference CTD data.

Our analysis of $\Delta S_{\text{Argo-refCTD}}$ shows that there are significant regional variations in the uncertainty of the Argo delayed-mode salinity dataset. In addition, there may be some residual bias that remains, possibly due to the difficulty in verifying small instrument calibration offsets in the absence of contemporaneous and co-located reference CTD data. These findings highlight several important points:

1. Even after delayed-mode evaluation and adjustment, some residual uncertainty can still remain in Argo salinity data. Historically, Argo's expected accuracy for salinity is 0.01 (Argo Science Team, 1998). This is not a metrologically-derived value, but is based on our experience, gained by data analysis (e.g. Riser et al., 2008; Wong et al., 2020), regarding the limitations of a delayed-mode system where data quality is assessed against sparse reference data and a changing ocean. Users should therefore take into account these residual uncertainties when using Argo delayed-mode salinity data.

2. There is a need for continual re-evaluation of the delayed-mode outcome against other independent references. These re-evaluation efforts need to be coordinated with the Argo delayed-mode community, and accompanied by collaborative efforts to update the data files and the relevant manuals to ensure common best practices.

3. Synergy between Argo and other ocean observing systems is vital in ensuring good data quality. Argo floats can provide good spatial and temporal coverage of the world's oceans, but high-quality reference data from independent platforms are needed to adjust and validate the data from floats.

4. Argo delayed-mode data can become available at different times and are subject to revisions as more reference data become available. Users should therefore refresh their data holdings periodically to obtain the most recent evaluation and adjustments.

**Data availability.** The Argo data used in this study are those available from the Argo Global Data Assembly Center in April 2022, https://doi.org/10.17882/42182#93132.

**Author contributions.** AW developed the concept for the manuscript, analyzed the data, wrote the manuscript, and produced the figures. JG compiled the data for analysis and contributed to the writing and discussions of the results. CC contributed to the writing and discussions of the results.

**Competing interests.** The authors have no competing interests to declare.

**Acknowledgements.** The authors wish to thank all the Argo delayed-mode operators for their work in improving this global dataset. Special thanks go to Christine Coatanoan for her work in maintaining the CTD_for_DMQC reference database. Comments from Birgit Klein and Mathieu Dever greatly improved the manuscript. Argo data are collected and made freely available by the International Argo Program and the national programs that contribute to it. Argo is part of the Global Ocean Observing System.

**Financial support.** AW was supported by the NOAA Global Monitoring and Observing Program via CICOES at the University of Washington through the project titled "The Argo Program - Global Observations for Understanding and Prediction of Ocean and Climate Variability". JG was supported by US Argo through NOAA Grant NA20OAR4320278 (CIMEAS/SIO Argo). CC was supported by the French National Centre for Scientific Research (CNRS).

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

**Short Summary (500 character non-technical text)**

This article describes the instrument bias in the raw Argo salinity data from 2000 to 2021. The main cause of this bias is sensor drift. Using Argo data without filtering out this instrument bias has been shown to lead to spurious results in various scientific applications. We describe the Argo delayed-mode process that evaluates and adjusts such instrument bias, and estimate the uncertainty of the Argo delayed-mode salinity dataset. The best ways to use Argo data are illustrated.