# Peer review of "Argo salinity: bias and uncertainty evaluation"

_Earth System Science Data, 2022_

## Author Response (AR1)

**Author's reply to Comment on essd-2022-323 by Birgit Klein**

General comments:

I find the manuscript well written and concise, and it gives a much needed evaluation of the biases and uncertainties in the Argo salinity corrections and the delayed mode data. It is a comprehensive assessment of the global data set and presents important findings about the magnitude of applied corrections, their spatial patterns and remaining differences to the reference data sets. I am sure this exercise is very helpful to inform the data user about the data set but also internally to address remaining questions, such as the maybe undetected minor fresh bias.

AW: Thank you to Dr. Klein for taking the time to review this manuscript, and for these positive comments.

Specific comments:

Page 2, line 38-39: I find the statement a bit misleading. The recovery of float was originally not intended and the floats were designed as 'throwaway products'. However, a recovery of floats is nowadays practised in marginal seas, where the distance from shore to float is small enough to reach the floats easy enough.

AW: This statement does not intend to suggest that float recovery is non-existent. Indeed several Argo groups have endeavoured to recover floats when opportunities arise, in the open oceans as well as in marginal seas. However, the number of such occasions is so small that we cannot rely on that method to recalibrate all floats. To clarify, I revised the sentence in L38-39 to: "Recalibration of these CTDs involves retrieval of the floats, which can occur when opportunities arise. However, such retrieval occasions are infrequent and not extensive. To determine if post-deployment adjustment of its data is necessary, Argo uses a set of delayed-mode procedures that makes use of reference data."

Page 2, line 46: 'sensor calibration drift' sounds odd, I would replace it by 'sensor drift'.

AW: Done.

Page 2, line 50: Is the term 'unadjusted salinity data' giving the right impression to the reader? The issued behind this is the use of data in R and A mode compared to D mode with scientific quality control in such analysis. Maybe a bit more text should be added.

AW: I revised L50 to: "The spurious increase in S after 2015 was postulated to be the result of inclusion of biased Argo salinity data that have not been adjusted in delayed-mode, while the absence ..."

Page 2, line 52: I would suggest to add a bit more information about the 'stricter quality control' in RG2009.

AW: The additional quality control in RG2009 is extensive. Thus it is better for the reader to refer to the paper itself.

Page 3, line 69: I would suggest to add a reference to section 2 for further information, about the +-0.01 uncertainty in quality controlled Argo salinities.

AW: I added Riser et al (2008) and Wong et al (2020) to L72 in the revision.

Page 4, line 106: maybe add 'at least yearly since the argo data reference data are updated more frequently.

AW: I added "approximately once a year" to L112 in the revision.

Page 4, line 118-119: the documents are available openly so maybe linking them to the Argo data management page http://www.argodatamgt.org/Documentation would be appropriate.

AW: The documents are cited by their DOIs, which is the appropriate way to cite references, because DOIs are unique and permanent identifiers. These are open access documents, and their DOIs point users to their open access landing pages. Webpages are not permanent and are therefore not the appropriate way to cite references. Moreover, the Argo data management webpage does not contain the documents, but merely lists the DOIs of the documents.

Page 5, line 128: 'dynamic data', I am not sure that the term is best to inform the reader that Argo data are potentially never reaching a final stage and could be subject to revisions even years after they have been calibrated and submitted in delayed mode.

AW: That is what "dynamic data" mean: "continually change and improve over time".

Page 5, lines 130-133: the paragraph about A-mode data might be lost to readers unfamiliar with the Argo data system and I am a bit unsure about its purposes. Do the authors want to discourage or encourage readers to use A-mode data?

AW: I agree this paragraph is out of place. I have moved the description of A-mode data to Section 4 in the revision.

Page 5, line 145: could the authors please give a range for salinity error in the strong gradient regions.

AW: I have added "can exceed 0.01 psu" to L146 in the revision.

Page 6, line 157: how do the authors identify pressure errors > 10 dbar in the profiles? Or do they mean when the PRESSURE_ADJUSTED_ERROR in the profile files is larger than 10 dbar.

AW: I have added "|Praw - Padjusted| > 10dbar" to L159 in the revision.

Page 7, line 192: 'signs of unrecoverable damage', maybe refer to the quality control manual for readers interested in more details.

AW: I have added the reference to the quality control manual to L199 in the revision.

Page 9: Figure 3, the highest corrections applied are from the Atlantic (mostly in the north) and North Pacific. Is there some explanation for the spatial patterns? Maybe the Atlantic pattern is caused by the fact that about 150 floats were purchased in a short time range when abrupt salty drift was most abundant and deployments were to a large extend carried out in the Atlantic. Maybe a version of figure 3 with data after 2015 could shed some light.

AW: The large magnitude corrections applied are mostly in float data before 2010, which are the earlier years of Argo when delayed-mode operators were less judicious in applying larger corrections. They are concentrated in the Atlantic and the North Pacific because in the earlier years of Argo, delayed-mode efforts were focused in those areas. I have added this explanation to L193-195 in the revision. I have also separated the overall mean in Figure 3 into two time blocks: (a) 2000-2010, and (b) 2011-2021.

Page 9: lines 209-214 to page 10: lines 215-224: I was wondering if such information about the Argo data structure in the files would not be needed earlier, but wouldn't know where to place it best.

AW: The information about Argo file structure is placed in Section 4, after the description of the biases in Section 3, so that the readers are first made aware of the differences between raw data and evaluated/adjusted data, and then introduced to how to find them in the Argo data files. This paper is more about describing the salinity bias and less about describing Argo file structures. The arrangement of the material in Sections 3 and 4 reflect this priority.

Page 12, line 242: I am not sure I am understand the purpose of this sentence about the greylist. Why would a user only use greylist to filter out suspicious data and not read the qc flags given in any float profile? Please check if this needs to be clarified.

AW: Relying on the grey list alone is not a good practice in Argo data usage, but it does happen, and hence this warning against this practice.

Page 13, line 275-277: It sound as if the main purpose of section 5 is to look for extremely bad data. But then in the following sentence it is explained that these gross

outliers are removed. As I understand the purpose of the section is to obtain statistical estimates of uncertainty in the delayed mode Argo data. Maybe change order of sentences or rephrase.

AW: I have changed the sentence to "it provides a means to examine the uncertainties in the global dataset". See L297 in the revision.

Page 13, line 284-288: the detailed description of the subtropical Atlantic where the most stable water masses are found in the central waters and not in the usual deep water range 1000-2000 gives it a strange importance. Could this be stated more generally, that even though deep layers are often the best choice there might be areas were that is not the case.

AW: This point is stated in Owens and Wong (2009). I have added this reference to L306 in the revision.

Page 13, line 291- 292: a search radius with a time window of 15 years might be problematic when there are temporal trends in the water masses. I would think that should be mentioned somewhere. And do I understand it correctly, that only 'the nearest' profile is used for comparison (i.e only 1 profile).

AW: I have added: "The comparison results will contain effects of spatial and temporal variabilities of the water masses, but there are minimized by using isotherms with relatively uniform salinity." to L319-321 in the revision.

Page 13, line 293: states that the differences are computed on the selected isotherms. In line 281 it sounds as if only one isotherm is selected in each 10x10 ° square. Does the plural just mean that the isotherms could have different value in each grid?

AW: Yes. I have clarified this in L315-316 in the revision.

Page 13, line 295: the fact that only 20% of the Argo profiles have nearby reference data with a search frame is a limitation of the analysis but also to the original dmqc analysis. Maybe some sentences should be added to the gaps in our existing reference data and a plea should be made to release CTD data more timely for the purpose of Argo data quality.

AW: The Argo DMQC analyses are not always restricted by a 3° radius circle. Moreover, Argo DMQC analyses are aided by a second ref dbase (Argo_for_DMQC). Hence the limitation of the analysis in Section 5 does not necessarily translate to limitation in Argo DMQC analyses. The point that high-quality reference data are needed to calibrate/validate Argo data is made in (3) in Section 6.  I have added two more sentences about the need for more reference CTD data in L344-347 in the revision.

Page 14, line 312-313: What about the ACC as a high uncertainty area? Or is that meant by South Atlantic?

AW: Circumpolar Deep Water (CDW) in the Southern Ocean, where the ACC is located, is a water mass with relatively uniform salinity. Argo DMQC analyses of float data in the Southern Ocean that use CDW as the reference usually produce robust results. Thus the Southern Ocean does not show up as a high uncertainty area in the analysis in Section 5. I have added this to L342-343 in the revision.

Page 15, line 316: figure 6, what is the meaning of the grey and white boxes? The colorbar used for figure 6 is a bit unusal for a difference plot and creates an optical bias towards the positive differences. For example the range -0.005 to 0 depicted by the light blue creates less attention than the similar 0 to +0.005 range depicted in light orange. I would suggest using a red/blue colour bar as in climate change maps. It would also be nice to have a plot, which indicates how many matches between climatology and Argo profiles went into each 10x10 square.

AW: I have changed the color scheme in Figure 6 and added an extra plot (Fig. 6b) that shows the number of Argo-refCTD match in each 10x10 square.  White are boxes with no Argo data or no Argo-refCTD match.

Page 15, lines 318-328: theses lines address the skewed distribution of differences with a bias to fresh and their spatial distribution. This is attributed to mild instrument calibration errors and the problems of veryfing this without contemporary reference data at deployment. Why should this only apply for fresh calibration offsets and not also for mild salty offsets?

AW: I have changed "mild fresh instrument offsets ..." to "mild instrument offsets, fresh or salty, ..." in L360 in the revision.

**Author's reply to Comment on essd-2022-323 by Mathieu Dever**

General Comments:

I believe this work to be very important. The Argo literature can sometimes be a bit scattered and overwhelming in volume. A manuscript summarizing the state of the Argo salinity database, and its limitations is a welcome contribution as it can guide new, and experienced, users on how to best use the data, and most importantly understand its limits.

The findings presented in this study are almost undersold: this is a valuable stepping stone towards improving the delayed-mode approach by recognizing the requirements and current limitations.

AW: Thank you to Dr. Dever for taking the time to review this manuscript, and for these positive comments.

Specific comments:

P1L27: I particularly appreciate this sentence "These results reinforce the need for continual re-evaluation of this global dataset." As I believe to be a crucial component of the Argo program.

AW: Thank you.

P3L68-69: The manufacturer's specs are misleading in salinity unit. As the CTD is calibrated in conductivity space, the conductive spec should only be stated. Does that salinity spec include the uncertainty on T and P propagated through the equation of state? Is it at Standard Temperature and Pressure (STP)? I think this might be better formulated in the way it is done in the conclusion: that the 0.01 accuracy is not from a metrological standpoint, but rather based on decades of data analysis.

AW: For completeness, it is useful to quote the manufacturer specified accuracies. I have added that 0.0035 psu corresponds to 0.0003 Siemens per meter at 2°C and 2000dbar, in L71 in the revision.

P5L146: As a setup for the next paragraph (i.e., only using >600 dbar data), I would maybe add a sentence here saying that because floats park at depth before profiling upward, they start their profiling phase mostly thermally equilibrated and thus only near-surface large temperature gradients are likely to affect the data quality, only propagating into the surface layer as the float is profiling upward.

AW: This is not quite true. It is true that the CTD is at thermal equilibrium during the drift phase at park depth. But for most floats, their profiling depth is deeper than park depth,

and so the floats need to descend to the profiling depth before ascending and sampling on the way up. Hence the CTD is not really thermally equilibrated at the start of the profiling phase.

Fig3: I'm slightly confused by what that figure shows. Is it the average, median, or maximum DM salinity adjustment per box? That stat might be biased by the number of float in the box. I am also noticing that the distribution of the salty drift match the density map of available reference profile. In other words, the more reference data we have available, the larger the drift is. If that is true, it could mean that some of the drift is missed due to a lack of available reference data.

AW: I have indeed forgotten to explain what Fig. 3 is. Thank you for catching that! The plot shows the overall mean of dS in each box. I have added that to the figure caption for Figure 3 in the revision. The mean is influenced by the distribution of dS in each box. In replying to the open comments from Reviewer 1, I discussed that the larger magnitude corrections are related to the earlier years of Argo, when delayed-mode operators were less judicious in applying larger corrections. They are concentrated in the Atlantic and the North Pacific because in the earlier years of Argo, delayed-mode efforts were focused in those areas, partly because those areas have more reference data, as you pointed out. I have added this to L193-195 in the revision. I have also separated the overall mean in Figure 3 into two time blocks: (a) 2000-2010, and (b) 2011-2021.

P10L219-219: This sentence sounds odd to me. Maybe something like "The raw data (DATA_MODE = 'R') can sometime be evaluated/adjusted in real-time (DATA_MODE = 'A', if available) before being evaluated/adjusted in delayed-mode (DATA_MODE = 'D'; See Fig 4)." I would also suggest maybe including an example of real-time adjustment? It might not be obvious to the reader, as the text is especially focused on sensor drift and that would never be a real-time adjustment.

AW: I have re-written the description of 'A' mode data. It is now in L228-231 in the revision.

P10L231-232. The point that BGC data is a whole different beast is actually quite important, and I would recommend including the Bittig et al (2019) reference and emphasizing that point in the introduction.

AW: I have included Bittig et al (2019) in L220 in the revision.

Table 1: QC flag '5' is a bit enigmatic to me. What could "value changed" mean? Maybe include an "e.g." in the box, or extra information in the table caption?

AW: The Argo QC flag table has an accompanying set of notes that provides additional information on the flags, including QC '5'. I have added that reference in the table caption.

P12L239: typo; should be "mean" and not "means".

AW: "means" here is used as a singular noun, referring to a method.

P12L240-241: Is inactivity the only reason why floats are removed from the grey list - or can they be removed on the operator's decision that the float is -after evaluation- not malfunctioning? If the latter, then I suggest to rephrase the sentence as it might be misleading as is.

AW: The latter can happen, but it is not common, and is not the point here. The point here is that the grey list is not a comprehensive list of problematic floats. I have clarified this point in L258-260 in the revision.

P14L308: The third and fourth moment of a PDF are not necessarily intuitive metrics, I think it would be helpful to remind the reader what they mean (positive skewness meaning longer tail on the positive â  S, and kurtosis>3 meaning heavy tails with large outliers.

AW: This is a good suggestion. Thank you. I have included a reminder of skewness in L349 and kurtosis in L336 in the revision.

P14L309: I am surprised by the large number of datapoints still lying outside the ±0.015 after delayed-mode adjustment (>18%). I think a discussion should be included on how to improve this number. The authors highlight a series of limitations to the method, but I think it would benefit from suggestions on how to improve the robustness of this very helpful analysis, which is a crucial point in monitoring the quality of the Argo fleet.

AW: I have included some discussion for more reference CTD in L344-347 in the revision.

P17L375-378: I believe this is a very important point and am glad to see it explicitly laid-out in this study!

AW: Thank you.